# Examining the Effects of the RUNX1 p.Leu43Ser Variant on FPD/AML Phenotypes Using a CRISPR/Cas9-Generated Knock-In Murine Model

**DOI:** 10.3390/biom15050708

**Published:** 2025-05-12

**Authors:** Ana Marin-Quilez, Ignacio García-Tuñón, Rocío Benito, José Luis Ordoñez, Lorena Díaz-Ajenjo, Ana Lama-Villanueva, Carmen Guerrero, Jesús Pérez-Losada, José Ramón González-Porras, Jesús María Hernández-Rivas, Mónica del Rey, José María Bastida

**Affiliations:** 1Cancer Research Center-CSIC, Instituto de Investigación Biomédica de Salamanca (IBSAL), University of Salamanca, 37007 Salamanca, Spain; ana.marin.94@gmail.com (A.M.-Q.); ignacio.tunon@uah.es (I.G.-T.); jlog@usal.es (J.L.O.); lorenadiaj21646@usal.es (L.D.-A.); ana29lv@usal.es (A.L.-V.); cguerrero@usal.es (C.G.); jperezlosada@usal.es (J.P.-L.); jmhr@usal.es (J.M.H.-R.); mdelrey@usal.es (M.d.R.); 2Servicio de Hematología y Oncología Médica, Hospital Universitario Morales Meseguer, Centro Regional de Hemodonación, University of Murcia, IMIB-Pascual Parrilla, CIBERER-U765, 30003 Murcia, Spain; 3Departament of Biomedicine and Biotecnology, Universidad de Alcalá, 28801 Alcalá de Henares, Spain; 4Laboratory of Pharmacology, Department of Physiology and Pharmacology, Faculty of Pharmacy, University of Salamanca, 37007 Salamanca, Spain; 5Department of Hematology, Complejo Asistencial Universitario de Salamanca (CAUSA), Instituto de Investigación Biomédica de Salamanca (IBSAL), Universidad de Salamanca (USAL), 37007 Salamanca, Spain; jrgp@usal.es

**Keywords:** *RUNX1*, myeloid neoplasm, FPD/AML, RNA-seq, murine model

## Abstract

Germline heterozygous variants in *RUNX1* lead to Familial Platelet Disorder with Myeloid Leukemia Predisposition (FPD/AML). Cellular and/or animal models are helpful to uncovering the role of a variant in disease progression. Twenty-five mice per genotype (RUNX1^WT/WT^, RUNX1^WT/L43S^, RUNX1^L43S/L43S^), previously generated by CRISPR/Cas9, and nine sub-lethally irradiated mice per genotype were investigated. Peripheral blood (PB), bone marrow (BM), and spleen samples were analyzed by flow cytometry and histopathology. Deregulated genes were analyzed by RNA-seq in BM. An aberrant myeloid Mac1^+^Sca1^+^ckit^−^ population in the PB, BM, and spleen of two homozygous and one heterozygous mouse was observed, as well as BM hypercellularity. No Mac1^+^Sca1^+^ckit^−^ cells were detected in any RUNX1^WT/WT^ mice. Moreover, the spleen of both homozygous mice showed destruction of the white/red pulp and the presence of apoptotic cells. The aberrant population was also detected in four irradiated mice, two heterozygous and two homozygous, in their PB, BM, and spleen. RNA-seq studies showed 698 genes significantly deregulated in the three non-irradiated Mac1^+^Sca1^+^ckit^−^ mice vs. six healthy mice, highlighting the alteration of genes involved in apoptosis and DNA repair. These results indicate that the homozygous form of the variant p.Leu43Ser may contribute to the pathogenesis of aberrant cells.

## 1. Introduction

The runt-related transcription factor 1 (*RUNX1*) gene encodes a transcriptional factor that is crucial in hematopoiesis and is mutated in ∼10% of adult acute myeloid leukemia (AML) cases [1,2]. Even though *RUNX1* variants are frequently diagnosed as an acquired alteration, the increased use of high-throughput sequencing (HTS) techniques in clinical practice has revealed that they are more prevalent than previously thought [3,4,5]. In the 2016 World Health Organization (WHO) classification, germline monoallelic alterations in *RUNX1* had been incorporated as a new subgroup: Familial Platelet Disorder with a predisposition to Acute Myeloid Leukemia (FPD/AML) [6,7]. It is estimated that ∼40% of individuals diagnosed with FPD/AML will develop a myeloid neoplasm, since germline *RUNX1* variants alone are insufficient to induce AML, and that second-hit mutational events are necessary [8,9]. The most frequent somatic mutations associated with leukemic progression have been described in the second allele of *RUNX1* but also in genes such as the cell cycle regulator *CDC25C*, the transcriptional factor *GATA2*, or genes associated with clonal hematopoiesis of indeterminate potential (CHIP), like *ASXL1* or *TET2* [10,11].

Therefore, an accurate and early diagnosis of these patients is critical for their prognosis and appropriate clinical management. Patients with FPD/AML require bone marrow examination and close follow-up for early recognition of a transformation to neoplasm and subsequent inclusion in a hematopoietic progenitor cell transplantation program [9].

The recent introduction of the Variant Curation Expert Panels has provided valuable guidance in categorizing *RUNX1* variants and in carefully monitoring patients for genetic counseling [12,13]. However, the application of these recommendations is still challenging, and there may be alternative interpretations or high controversies for many genetic variants [14]. The development of genetically modified murine models and the use of induced pluripotent stem cells (iPSCs), in vitro models, and/or transcriptomic analysis have proven to be a suitable model system to study genetic disorders, providing new insight into FPD/AML [15,16,17,18].

Our group previously published the generation of the RUNX1 p.Leu43Ser murine model generated by CRISPR/Cas9, reproducing the human RUNX1 p.Leu56Ser variant, to characterize the related platelet disorder [19]. In this study, we examine the role of this variant by identifying an abnormal cell population in two homozygous mice and one heterozygous mouse carrying the variant. Furthermore, these mice demonstrate atypical expression of genes associated with apoptosis and DNA repair.

## 2. Materials and Methods

### 2.1. Ethics Statement

Animal studies were conducted in accordance with the Spanish and European Union guidelines for animal experimentation (RD53/2013 and Directive-2010/63/UE, respectively) and received prior approval from the Bioethics Committee of the University of Salamanca and the Junta de Castilla y León, Spain (0000107).

### 2.2. Murine Model and Experimental Design

A murine model mimicking the human RUNX1 p.Leu56Ser variant, previously generated by CRISPR/Cas9 technology, was used [19]. A total of 75 mice were investigated: 25 of each genotype (RUNX1^WT/WT^, RUNX1^WT/L43S^, and RUNX1^L43S/L43S^) (50% of each gender). Also, nine mice per genotype (6 to 8 weeks old, 41% female, 59% male) received 4 Gy of total-body γ-irradiation from a 137Cs source. The mice were housed at the Servicio de Experimentación Animal (SEA), University of Salamanca (Salamanca, Spain), a temperature-controlled specific-pathogen-free animal house facility.

All mice were followed daily, and blood cell populations were screened routinely by flow cytometry (FC) every 3 months or upon suspicion of myeloid disease. Mice were sacrificed at the end of the experiment (22–24 months) or when presenting symptoms of illness, as described [20]. Peripheral blood (PB), bone marrow (BM), and spleen samples were collected at sacrifice.

### 2.3. Characterization of Hematopoietic Cells by Flow CytometryPanel by Thermo Fisher

Blood samples (100–200 µL) were collected from the submandibular vein of adult mice and anticoagulated with EDTA for routine screening. At sacrifice, PB was collected by cardiac puncture of anesthetized mice (tribromoethanol), BM cells were obtained by flushing from the long bones, and spleen cells were obtained by mechanical procedures. In all cases, contaminating red blood cells were lysed with red cell lysis buffer (RCLB: NH_4_Cl, KHCO_3_, EDTA), and the remaining cells were washed in PBS for flow cytometry, as described [20]. Cells were stained with a customized panel: Gr1*FITC (1:100); Mac1*PE (1:200); CD45*PerCP-Cy5.5 (1:100); ckit*PECy7 (1:50); CD3*APC (1:100); B220*APCH7 (1:100); Sca1*PacificBlue (1:50) (Biolegend, San Diego, CA, USA). Samples were incubated for 30 min at room temperature (RT), in the dark, and acquired in a BD FACSAria Cytometer (BD Biosciences, Milpitas, CA, USA). Data were analyzed using FlowJo V10 (Tree Star, Inc., Ashland, OR, USA).

### 2.4. Histopathology

Paraformaldehyde-fixed femur and spleen tissue samples were immersed in paraffin and cut into 2 μm thick slices. Blood films were fixed with ethanol. All samples were stained with hematoxylin–eosin. Representative samples were photographed under an Olympus BX51 microscope connected to an Olympus DP70 camera (Olympus, Tokyo, Japan). The histopathological diagnosis was carried out by experts from The Molecular Pathology Unit (Salamanca, Spain).

### 2.5. RNA-Seq from Bone Marrow Samples

Leukocytes from mouse bone marrow were resuspended in RLTplus lysis buffer from an RNeasy kit (Qiagen, Hilden, Germany). Total RNA extraction was obtained following the manufacturer’s specifications.

RNA-seq and bioinformatic analyses were performed at the Sequencing Service of NUCLEUS, University of Salamanca. Raw fastq files were first quality-filtered using fastp (v0.23.2), and processed fastq files were aligned to the mouse genome (mm10) using the STAR aligner (v2.7.8a). FeatureCounts (v1.50) was used to summarize the reads across the genes, using the vM25 version of the comprehensive GENCODE mouse gene annotation. A differential expression analysis between the experimental groups was performed using DESeq2 (v2.11.40.6). Lastly, the goseq (v1.44.0) enrichment analysis package was used to conduct a gene ontology analysis of differentially expressed genes [21,22].

### 2.6. Statistical Analysis

Data were summarized as the mean ± standard error of the mean (SEM). Statistical analyses were performed using GraphPad Prism 8 Software (GraphPad Software, San Diego, CA, USA). Differences among groups were tested with t-student, one- and two-way ANOVA, and Tukey’s multiple comparisons test. Differences were considered significant (*) for values of *p* < 0.05.

## 3. Results

### 3.1. Phenotyping of Aberrant Cells in Peripheral Blood, Bone Marrow, and Spleen

To study the effect of the RUNX1 p.Leu43Ser variant on both mature and immature hematopoietic populations in PB, the mice were followed throughout their lives. We observed normal counts of red blood cells; a congenital reduction in the platelet count, as previously described [19]; and an allele-dependent reduction in leukocyte counts in PB (Figure 1A) without changes in the expected proportions of mature cells (B and T lymphocytes, granulocytes, and monocytes) at sacrifice (Figure 1B). Regarding immature cells, no Lin^−^ ckit^+^ sca1^+^ (LSK) cells were detected in PB. However, we identified an aberrant population characterized by being double-positive for the markers Mac1 and Sca1, and negative for ckit (Mac1^+^ Sca1^+^ cells), in three mice: two RUNX1^L43S/L43S^ (referred to as Hom#1 and Hom#2) and one RUNX1^WT/L43S^ (referred to as Het#1) (Figure 1C). Mouse Hom#1 displayed 13.4% of Mac1^+^ Sca1^+^ cells at 14 months and 15.3% of cells at sacrifice (15 months). Conversely, mouse Hom#2 displayed a milder and later appearance of the aberrant population (7.3% at 20 months). Finally, only one heterozygous mouse, Het#1, presented the Mac1^+^ Sca1^+^ population at an advanced age (21 months: 2.4% of cells) (Figure 1C). A blood smear revealed the presence of cells with a morphology characterized by the absence of nucleoli, dysmorphia in the nucleus, and a small cytoplasm, which could potentially be the aberrant population identified (Figure 1D). Moreover, the Mac1^+^ Sca1^+^ aberrant cells had similar fluorescence intensity to that of monocytes for the CD45 marker, but they were smaller and less complex, based on the FSC-SSC distribution (Figure 1E).

Regarding hematopoietic organs, there were no significant differences in LSK levels (Lin^−^ Sca1^+^ ckit^+^) or progenitors (Lin^−^ Sca1^−^ ckit^+^) between RUNX1^WT/L43S^, RUNX1^L43S/L43S^, and RUNX1^WT/WT^ mice in their BM (Figure 2A). However, the three affected mice also presented high levels, above average, of both stem cells and progenitor populations in their BM (Figure 2A). Aberrant Mac1^+^ Sca1^+^ cells were also found in these mice (Figure 2A). This aberrant population has a differentiation stage that is intermediate between LSK cells and mature monocytes (Figure 2B). The anatomo-pathological study revealed BM hyperplasia in mice Hom#1, Hom#2, and Het#1 compared with RUNX1^WT/WT^ mice but also with unaffected RUNX1^WT/L43S^ or RUNX1^L43S/L43S^ mice (Figure 2C).

A similar phenotype was found in the spleen of the affected mice Het#1, Hom#1, and Hom#2, characterized by a substantial increase in both stem cells and progenitor populations, as well as the appearance of the Mac1^+^ Sca1^+^ population (Figure 3A). As shown in Figure 3B–E, the RUNX1^WT/L43S^ mice did not exhibit spleen abnormalities, including the Het#1 mouse. In contrast, the RUNX1^L43S/L43S^ mice displayed variability in their spleen size and weight (Figure 3B,C). Although the spleen size in the RUNX1^L43S/L43S^ mice showed an increase (*p* = 0.08) (Figure 3B), notable splenomegaly was only observed in the mouse Hom#2 (Figure 3C). Nevertheless, Hom#1 displayed remarkable tissue structuring impairments, characterized by the presence of necrotic and apoptotic cells (Figure 3D,E).

### 3.2. Irradiation Increases the Risk of Aberrant Cell Proliferation in Mice Carrying the RUNX1 p.Leu43Ser Variant

Nine mice per genotype were irradiated at sub-lethal doses to promote the development of second events leading to leukemogenesis. Similar to the findings obtained in non-irradiated mice, we detected in the PB similar proportions of mature populations (B and T lymphocytes, granulocytes, and monocytes) among the irradiated mice of different genotypes at sacrifice (Figure 4A).

No LSK cells were observed in PB, but we detected the presence of the aberrant Mac1^+^ Sca1^+^ population in two heterozygous and two homozygous mice. Specifically, mouse I-Het#1 presented 7.2% of Mac1^+^ Sca1^+^ cells at 19 months and 13.8% of these cells at sacrifice (21 months), while I-Het#2 displayed a later appearance of the aberrant population (16.8% at 24 months) (Figure 4B). Additionally, I-Hom#1 was sacrificed at 9 months with 2.3% of aberrant cells, while I-Hom#2 had 11.5% of Mac1^+^ Sca1^+^ cells at 24 months (Figure 4B).

No significant differences in LSK cells or progenitors (Lin^−^ Sca1^−^ ckit^+^) were observed between mice of different genotypes in the BM or spleen (Figure 4C,D), but I-Het#1, I-Het#2, and I-Hom#2 presented high levels, above average, of both immature cells. Furthermore, aberrant Mac1^+^ Sca1^+^ cells were found in these three mice in both the BM and spleen (Figure 4C,D).

### 3.3. RNA-Seq Revealed Aberrant Gene Expression in Hom#1 and Hom#2 Mice

RNA-seq was performed to investigate possible pathways and genes deregulated in the affected mice that could explain the phenotype. Inthe initial study, a comparative analysis was conducted on three non-irradiated, unaffected mice carrying the variant (one RUNX1^WT/L43S^ and two RUNX1^L43S/L43S^) versus three non-irradiated RUNX1^WT/WT^ mice. The results of this analysis revealed only five dysregulated genes (*Gm16698, Vκ21G, Rbm44, Gzma*, and the LncRNA 5830416I19Rik). These findings indicate that the variant p.Leu43Ser does not, in itself, cause the observed phenotype. Therefore, these six mice were grouped in the RNA-seq analysis as a group named “healthy mice” and were compared with the non-irradiated affected mice with the aberrant population Mac1^+^ Sca1^+^ (Hom#1, Hom#2, and Het#1 mice), revealing 698 deregulated genes.

An enrichment analysis revealed that the deregulated genes were mainly involved in apoptosis and DNA repair pathways (Figure 5A), but affected genes related to hypoxia, glycolysis, and inflammatory response, among others, were also detected (Appendix A). Moreover, using the Wallenius method, the top over-represented categories revealed that the most disrupted pathways were related to changes in protein binding and immune response (Appendix A).

The top 50 deregulated genes are shown in Figure 5B and Appendix A. Twelve target genes were deeply investigated according to the profile of expression of the three affected mice vs. the six control mice. Among the downregulated genes in the affected mice with Mac1^+^ Sca1^+^ aberrant cells, we observed *Klra1*, a membrane receptor expressed mainly on the surface of NK cells and other cells of the immune system; *Fcer2a*, involved in the humoral immune response; *Ppp1r42*, a regulator of centrosome separation; *Rhoc*, a member of the Rac subfamily of the Rho GTPases; *Bnip3*, a gene that interacts with anti-apoptotic proteins, including BCL2; *Tmem38a*, a cation channel required for the maintenance of rapid intracellular calcium release; and *Cr2*, which codifies for receptor 2 of the complement C3d protein. Otherwise, the overexpressed genes in the three affected mice with the aberrant cells included *Parm1*, a mucin-like androgen-regulated gene; *Rapgef1* (also known as C3G), a guanine nucleotide exchange factor that activates Rap1 GTPases, with a relevant function in platelet hemostasis [23,24,25]; *Dnajb13*, involved in sperm terminal differentiation; *Gpr55*, a G-protein-coupled receptor superfamily that regulates proliferation; and *Slco2a1*, a prostaglandin transporter.

Overexpression of *Rapgef1* has been previously associated with several hematopoietic malignancies [26,27]. In fact, we found in our affected mice that the more aggressive phenotype was associated with increased expression of *Rapgef1*, since the Hom#1 and Hom#2 mice presented a higher overexpression of the gene than did the Het#1 mouse (Figure 5B and Appendix A).

## 4. Discussion

The landscape of FPD/AML evolution is complex and heterogeneous. Germline *RUNX1* variants per se are enough to cause a platelet disorder. However, the molecular evolution in clonal hematopoietic cells that leads to myeloid malignancy development remains poorly understood [28,29], since a second-hit mutational event is required for leukemogenesis [30]. The most frequent second event in FPD/AML patients is the somatic mutation of the unaffected allele of *RUNX1* [10]; however, mutations in other genes such as *CDC25C*, *CBL*, *FLT3*, *TP53*, or *ASXL1*, among others, have also been described [31,32]. Moreover, the increasing use of genome editing tools has allowed the development of cellular or animal models reproducing the *RUNX1* variants to characterize the associated mechanisms of pathogenicity [33,34,35]. In 2021, Decker et al. published that RUNX1 p.Leu56Ser has normal dimerization with CBFβ and RUNX1 phosphorylation but reduced transcriptional activation of genes such as r*ETV1* and r*CSF1R* [15]. Furthermore, Koh et al. described the inability of RUNX1 p.Leu56Ser to bind MLL, suggesting a novel model of leukemogenesis related to the variant [36]. All these results, together with the variability of in silico prediction tools classifying the variant as benign but also pathogenic, bring into focus the functional classification of the variant as one of uncertain significance. However, given the high penetrance of the variant in heterozygosis in the population (0.0188–GnomAD exomes/0.0126–GnomAD genomes), and following the rules of the ACMG MM-VCEP, this variant should be classified as benign [13].

Recently, our group described the optimal generation of a murine model carrying the RUNX1 p.Leu43Ser variant, generated by the CRISPR/Cas9 tool, and mimicking the human p.Leu56Ser. Heterozygous but especially homozygous mice displayed impaired platelet function characterized by decreased α_IIb_β_3_ integrin activation, aggregation, and PKC α/β phosphorylation, demonstrating the role of the variant in platelet disorders [19]. Therefore, to settle the role of the variant in leukemic progression, we evaluated the mature and immature hematopoietic populations throughout life. We reported the appearance of an aberrant population in PB that was double-positive for the markers Mac1 and Sca1 and negative for ckit, previously described as myeloid leukemic cells [37], in two homozygous mice (Hom# and Hom#2). This aberrant population appeared earlier (15 months vs. 20 months) in Hom#1, with a higher percentage of aberrant cells than in Hom#2 (Figure 1C), demonstrating the significant heterogeneity that exists even among mice with the same genotype. In contrast, only one heterozygous mouse presented this aberrant Mac1^+^ Sca1^+^ population, at a remarkably advanced age (21 months, equivalent to 60–70 years in humans) and with relatively lower levels of the aberrant population than homozygotes. None of the WT displayed the aberrant population. These results suggest that the presence of the population is dependent on the p.Leu43Ser variant. Moreover, a BM analysis of the three mice showed increased levels of LSK cells, progenitors, and aberrant cells. No alterations in the spleen were found in the RUNX1^WT/L43S^ genotype, while RUNX1^L43S/L43S^ mice had a major tendency to splenomegaly, especially in the Hom#2 mouse. It has been previously described that the levels of *RUNX1* activity are critical for leukemic predisposition [38], justifying the Sca1^+^ Mac1^+^ cells that were found in our two RUNX1^L43S/L43S^ mice but much less significant in RUNX1^WT/L43S^. Nevertheless, the overall incidence of myeloid neoplasm related to this variant was remarkably lower than that described for *RUNX1*-mutated patients [39], since only 4% of the heterozygous mice (1/25) and 8% of the homozygous mice (2/25) developed the aberrant phenotype, compared to the 44% leukemic progression in FPD/AML patients described in the literature [1]. Thus, these results suggest that RUNX1 p.Leu43Ser in heterozygous form is associated with a very low risk of leukemogenesis (4%), similar to findings in patients with RUNX1 p.Leu56Ser, but is also related with advanced age. In contrast, RUNX1 p.Leu43Ser in homozygous form is related to increased penetrance and an earlier age of onset, but there is variability in the phenotypic expression. Furthermore, the homozygous variant was identified in 357 of the patients included in GnomAD exomes (MAF: 0.0002). Consequently, monitoring these patients should be a major consideration for genetic counseling.

In addition, studies in irradiated mice, in which we promoted the development of the second hit, showed a higher incidence of aberrant cell development in RUNX1^WT/L43S^ and RUNX1^L43S/L43S^ mice than in RUNX1^WT/WT^ mice (22%, 22%, and 0%, respectively). Although these results are not significant, we observe a trend, suggesting that the variant has a low predisposition to the development of malignant cells and that its frequency increases when animals are exposed to irradiation and, thus, to the development of second events.

To obtain significant results, a larger cohort of mice would be required. However, our study complied with the 3R principles of animal experimentation (reduce, replace, refine) as approved by the Bioethics Committee of the University of Salamanca.

In addition, to further characterize the mechanism underlying the appearance of the aberrant population in the three non-irradiated mice (Hom#1, Hom#2, and Het#1), RNAseq was performed in order to investigated deregulated genes. A transcriptomic analysis of different phenotypes (normal phenotype vs. aberrant phenotype) showed an important number of genes that were significantly deregulated (698). Within the significantly deregulated genes, we observed the deregulation of genes involved in cell cycle regulation, apoptosis, and DNA repair, but also in hypoxia, glycolysis, and inflammatory response. These results suggest that high-dose cytotoxic drugs should be avoided in this group of patients. Moreover, these results are consistent with those of recently published studies describing a novel mechanism of AML development involving elevated inflammatory responses, the mutation of *CXXC4*, and decreased *TET2* levels [40].

It is important to mention that *RUNX1* was not altered, neither transcriptionally nor molecularly, since all exons of the gene were studied in the mice with the aberrant population and none presented somatic mutations in the healthy allele of *RUNX1*, which is considered the second most frequent event in the development of FPD/AML. Most of the significantly deregulated genes are involved in different biological processes such as the regulation of GTPase activity. In our study, we observed the overexpression of *Rapgef1*. This guanine nucleotide exchange factor activates several members of the Ras superfamily of GTPases, mainly Rap1, the main GTPase that regulates platelet function, and its alteration has been described in several neoplasms, including hematological malignancies such as chronic myeloid leukemia [25,26]. In fact, the regulation of this gene by PKCα/β functional activity has been described [23,41].

Moreover, a correlation was found between the overexpression of *Rapgef1* and the genotype and phenotype, since the higher expression of this gene was more remarkable in RUNX1^L43S/L43S^ mice, which displayed a more aggressive phenotype. These results agree with those of previous studies demonstrating the importance of allelic burden in *RUNX1* and its relevance to FPD/AML progression [38], suggesting that *RUNX1* alterations could impair the PKCα/β signaling pathway, promoting the disease’s appearance. Normal levels with reduced function of PKC-α/β are implicated in platelet dysfunction, as previously published [19]. Meanwhile, during leukemic progression, the overexpression of *Rapgef1* could increase cell proliferation and myeloid neoplasm progression in our affected mice through hyperactivation of the Rap1 signaling pathway, as previously described [25,42,43]. However, these results should be confirmed by RT-qPCR, which could not be performed in this study, as well as expanded and confirmed by studying patients and more variants of the *RUNX1* gene. In this context, it would be of significant interest to examine the specific genes regulated by *RUNX1*, as the measurement of them could provide valuable insights into the pathogenicity of the variant.

## 5. Conclusions

Our murine model generated using CRISPR/Cas9 revealed the presence of abnormal cells in 8% of the homozygous forms of p.Leu43Ser, underscoring the importance of allele burden in *RUNX1*. These studies demonstrated the deregulation of genes that are essential for maintaining hematopoiesis, cell cycle regulation, DNA repair, and inflammation. However, further research is needed to establish the role of these genes, such as *Rapgef1*, in leukemogenesis.

## Figures and Tables

**Figure 1 biomolecules-15-00708-f001:**
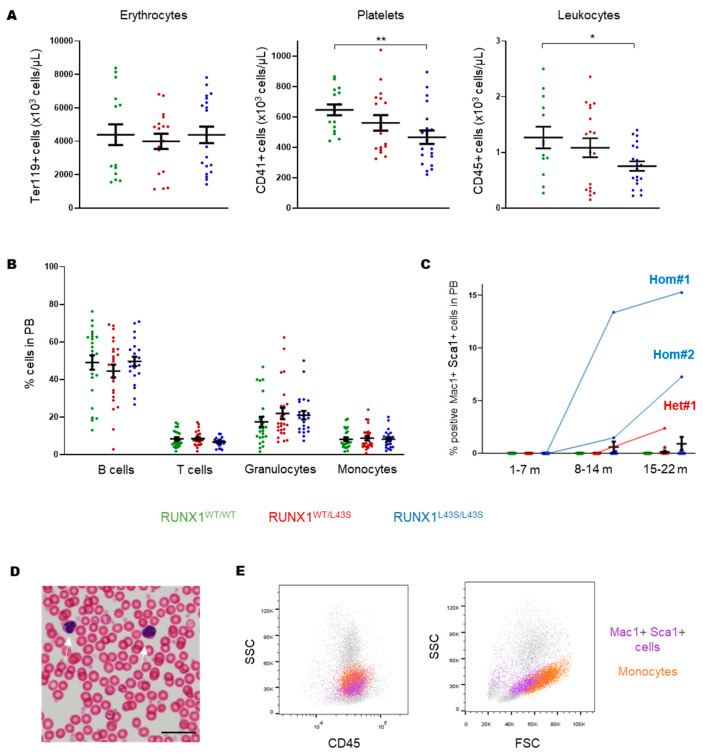
Characterization of hematopoietic cells in peripheral blood (PB) from RUNX1^WT/WT^, RUNX1^WT/L43S^, and RUNX1^L43S/L43S^ mice. (**A**) Erythrocyte, platelet, and leukocyte counts, measured by flow cytometry, using the monoclonal antibodies anti-Ter119, anti-CD41, and anti-CD45, respectively. The plots show the total number of cells/µL. (**B**) Leukocytes were stained with specific monoclonal antibodies to measure the percentages of B cells (anti-B220), T cells (anti-CD3), granulocytes (double-positive for anti-Gr1 and anti-Mac1), and monocytes (negative for anti-Gr1 and positive for anti-Mac1) in PB at sacrifice. (**C**) Leukocytes were stained with the monoclonal antibodies anti-Mac1 and anti-Sca1 to detect the aberrant population. The percentage of double-positive population is shown at different periods of life for RUNX1^WT/WT^, RUNX1^WT/L43S^, and RUNX1^L43S/L43S^ mice, including the three affected mice Hom#1, Hom#2, and Het#1. (**D**) The morphological appearance of the Mac1^+^ Sca1^+^ aberrant population in blood film, stained with hematoxylin–eosin, in the affected mouse Hom#1 (white arrows) Bar: 10 µm. (**E**) FCS/SSC and CD45^+^/SSC plots gating the Mac1^+^ Sca1^+^ aberrant population (purple) and the mature monocytes (orange) in the affected mouse Hom#1. Dot plots represent the mean ± SEM. * *p* < 0.05, ** *p* < 0.01. m: months.

**Figure 2 biomolecules-15-00708-f002:**
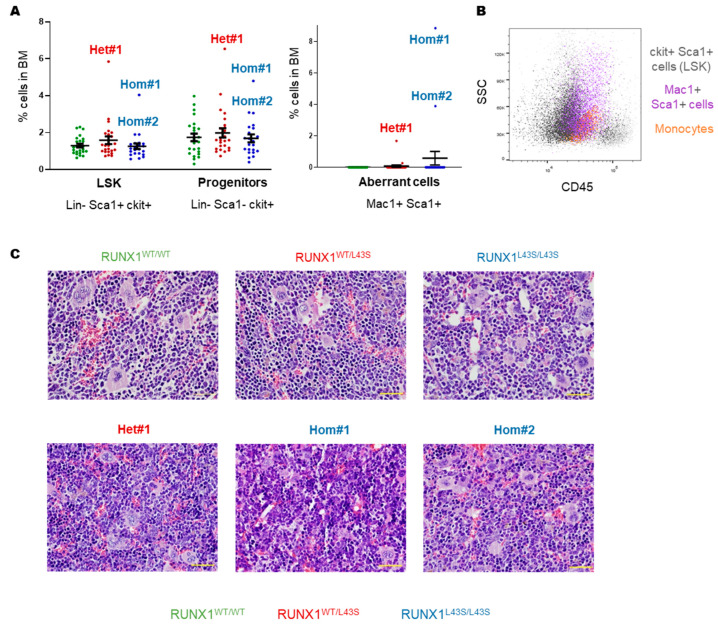
Characterization of the aberrant cell population in bone marrow (BM) from the Het#1, Hom#1, and Hom#2 mice. (**A**) Leukocytes from BM were stained with lineage-specific monoclonal antibodies (Lin: anti-B220, anti-CD3, anti-Gr1, and anti-Mac1) and specific monoclonal antibodies to analyze immature cells (Sca1 and ckit). The aberrant population was detected with the double-positive population Mac1 and Sca1. Dot plots represent the mean ± SEM of the percentage of the cell population. (**B**) CD45^+^/SSC plots gating the Mac1^+^ Sca1^+^ aberrant population (purple), the mature monocytes (orange), and the LSK cells (sca1^+^ ckit^+^; black) in the affected mouse Hom#1. (**C**) A histopathological assessment of BM anatomy after staining with hematoxylin–eosin. Bar: 100 μm.

**Figure 3 biomolecules-15-00708-f003:**
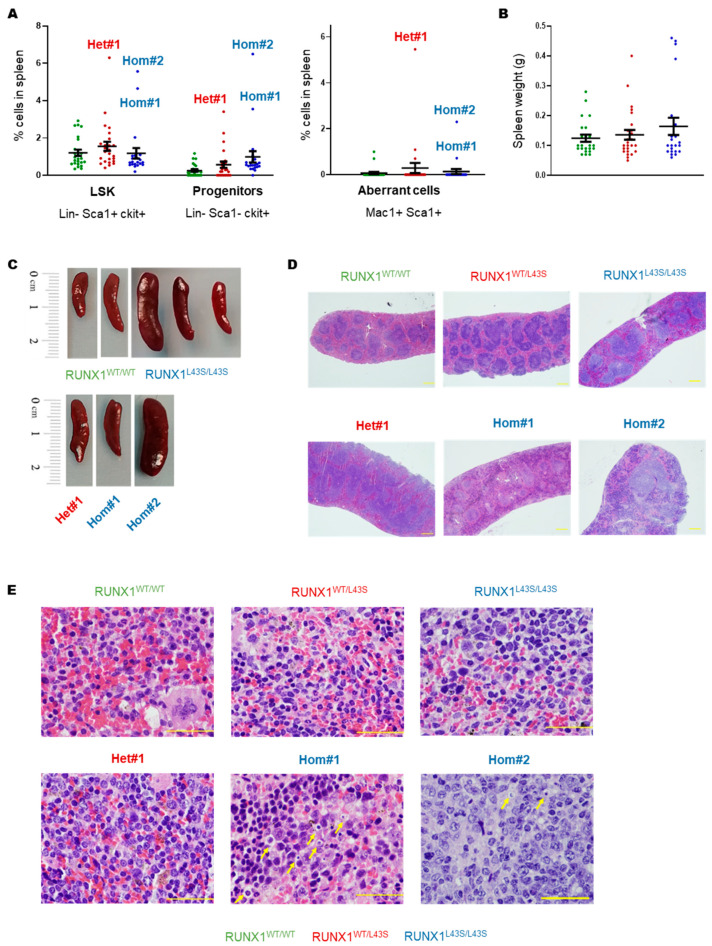
Characterization of the aberrant cell population in spleens from the Het#1, Hom#1, and Hom#2 mice. (**A**) Leukocytes from the spleens were stained with monoclonal antibodies for lineage-specific (anti-B220, anti-CD3, anti-Gr1, and anti-Mac1) and immature (Sca1 and ckit) populations. The aberrant population was detected with the double-positive population Mac1 and Sca1. Dot plots represent the mean ± SEM of the percentage of the cell population. (**B**) Splenomegaly was determined by the spleen weight (g). Dot plots represent the mean ± SEM of values per genotype. (**C**) A representative figure of the splenomegaly associated with RUNX1^L43S/L43S^ vs. RUNX1^WT/WT^ mice, and the spleens of the three affected mice. (**D**,**E**) A histopathological assessment of the spleen anatomy after staining with hematoxylin–eosin. The yellow arrows indicate apoptotic cells. Bar: 100 μm.

**Figure 4 biomolecules-15-00708-f004:**
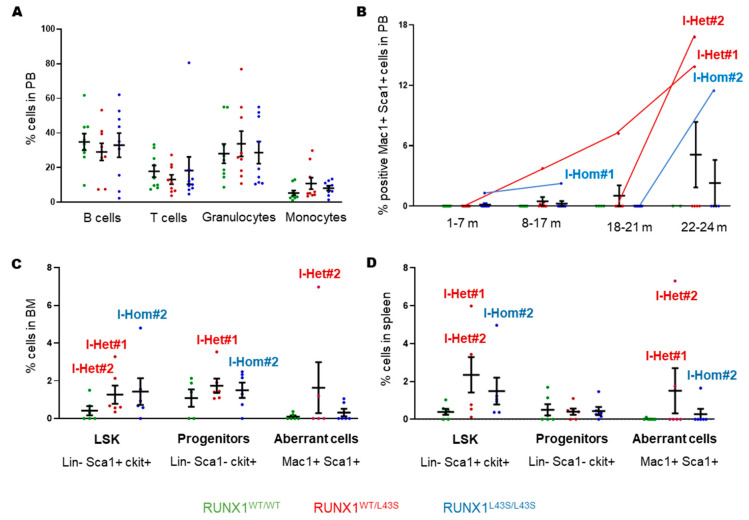
Characterization of hematopoietic cells in peripheral blood (PB), bone marrow (BM), and spleens from irradiated RUNX1^WT/WT^, RUNX1^WT/L43S^, and RUNX1^L43S/L43S^ mice. (**A**) Leukocytes were stained with specific monoclonal antibodies to measure the percentages of B cells (anti-B220), T cells (anti-CD3), granulocytes (double-positive for anti-Gr1 and anti-Mac1), and monocytes (negative for anti-Gr1 and positive for anti-Mac1) in the PB of irradiated mice at sacrifice. (**B**) Leukocytes were stained with the monoclonal antibodies anti-Mac1 and anti-Sca1 to detect the aberrant population. The percentage of the double-positive population is shown at different periods of life for irradiated RUNX1^WT/WT^, RUNX1^WT/L43S^, and RUNX1^L43S/L43S^ mice, including the four affected mice I-Hom#1, I-Hom#2, I-Het#1, and I-Het#2. (**C**,**D**) Leukocytes were stained with lineage-specific monoclonal antibodies (Lin: anti-B220, anti-CD3, anti-Gr1, and anti-Mac1) and specific monoclonal antibodies to analyze immature cells (Sca1 and ckit) in the (**C**) BM and (**D**) spleen. The aberrant population was detected with the double-positive population Mac1 and Sca1. Dot plots represent the mean ± SEM of the percentage of the cell population.

**Figure 5 biomolecules-15-00708-f005:**
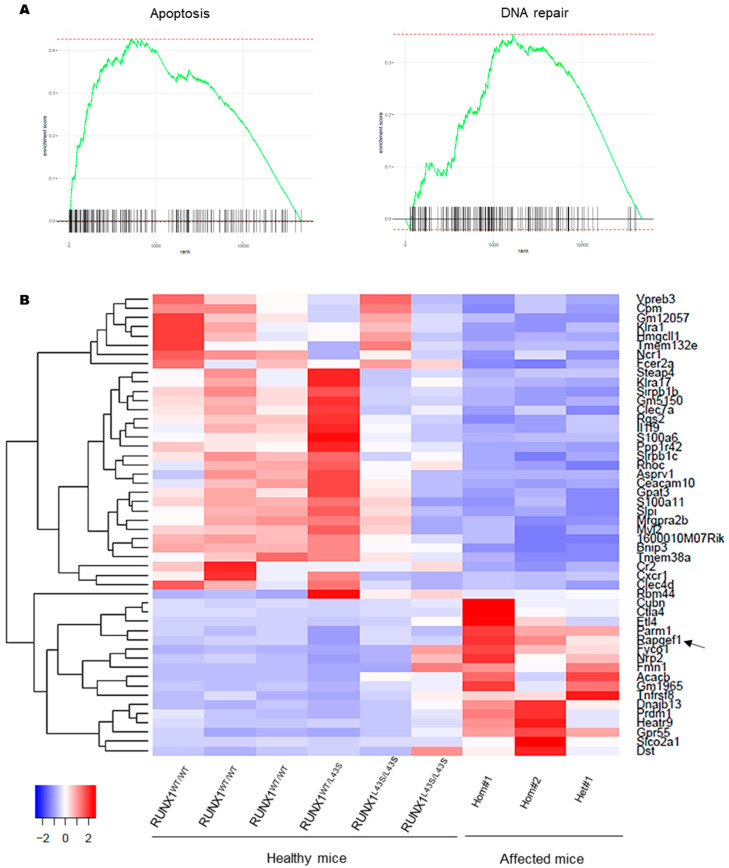
RNA-seq revealed altered expression of genes. (**A**) The enrichment scores revealed deregulated genes involved in apoptosis and DNA repair using FGSEA. (**B**) A heatmap of the top 50 differentially expressed genes in the three affected mice with Mac1^+^ Sca1^+^ aberrant cells. The heatmap was built using DESeq2 and Heatmap2 on normalized gene counts. The black arrow indicates the overexpression of Rapgef1 as a potential second-hit event.

## Data Availability

The original contributions presented in this study are included in the article/Appendix A. Further inquiries can be directed to the corresponding author.

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
