# Peer review of "Examining the Effects of the RUNX1 p.Leu43Ser Variant on FPD/AML Phenotypes Using a CRISPR/Cas9-Generated Knock-In Murine Model"

_biomolecules, 2025, doi:10.3390/biom15050708_

Round 1

Reviewer 1 Report

Comments and Suggestions for Authors

Overview

RUNX1 is a transcriptional factor that is mutated in FPD/AML in human. Here a mouse model carrying L43S mutation in Runx1 gene was investigated for the changes in peripheral blood cell populations, bone marrow and spleen. An aberrant leukocyte cell population (Mac1+, Sca1+, ckit-) was identified in peripheral blood, bone marrow and spleen in several homozygous and heterozygous mice. RNAseq revealed deregulation of multiple genes from apoptosis and DNA damage response pathways in total leukocyte population from these animals.

Major concerns

In my opinion, the presented data suggest that the RUNX1 mutation L43S, in fact, has little effect on the appearance of the identified aberrant population Mac1+ Sca1+ ckit –.

First, the experimental groups included 25 mice per each genotype, but the aberrant population Mac1+ Sca1+ ckit – was identified only in three non-irradiated mice (two homozygous for L43S and one heterozygous) and in four irradiated mice (2 x het, 2 x homo). I think these results are not statically compelling (1-2 mice out of 25). These data suggest that the presence of the aberrant cell population largely is independent of the L43S mutation. This is partly confirmed by the RNA seq results which showed dysregulation of multiple genes only in mice with aberrant populations, but not in all L43S/L43S animals.

Altogether, the presented data suggest that in animals with aberrant cell population a second mutagenic event has occurred that has led to the development of the observed phenotype, but the L43S mutation has little effect on it. This is partly confirmed by the lack of dramatic AML phenotype in homozygous mice versus wild type mice and by the fact that corresponding L56S mutation in human is considered benign (see discussion).

Another issue is that this second mutagenic event is unknown. Moreover, these second hits could be different in all mice with aberrant populations thus hampering the interpretation of results. Taking into account the lack of a dramatic phenotype in most of the studied animals one might assume that the development of the aberrant population depends on the second event, while Runx1 mutations only marginally contribute to this phenotype. This is exemplified by the higher percentage of aberrant population was higher in non-irradiated homozygous mice versus heterozygous (Fig1) suggesting that the L34S/L34S genotype makes mice prone to the development of this aberrant population, but still without second event there will be no aberrant population. I think that it is important to identify this second hit event in affected mice and study its relationship with Runx1 mutations.

Taking together the complex relationship between Runx1 mutations and possible second hit mutations should be investigated in greater details.

Minor issues

The most common second event that leads to the development of AML is the mutation of the second unaffected RUNX1 allele. But L65S mutation seems to be benign in human so why this particular mutation was selected for analysis?

The changes in mRNA levels of RNA-seq identified dysregulated genes should be confirmed in independent experiments using qPCR for these genes (e.g. for Rapgef1 gene).

Fig1AB-  Experimental points are not labeled (those that are shown in green, red and blue). Fig1 title should be in bold.

Author Response

Overview

RUNX1 is a transcriptional factor that is mutated in FPD/AML in human. Here a mouse model carrying L43S mutation in Runx1 gene was investigated for the changes in peripheral blood cell populations, bone marrow and spleen. An aberrant leukocyte cell population (Mac1+, Sca1+, ckit-) was identified in peripheral blood, bone marrow and spleen in several homozygous and heterozygous mice. RNAseq revealed deregulation of multiple genes from apoptosis and DNA damage response pathways in total leukocyte population from these animals.

 Major concerns

In my opinion, the presented data suggest that the RUNX1 mutation L43S, in fact, has little effect on the appearance of the identified aberrant population Mac1+ Sca1+ ckit–. First, the experimental groups included 25 mice per each genotype, but the aberrant population Mac1+ Sca1+ ckit – was identified only in three non-irradiated mice (two homozygous for L43S and one heterozygous) and in four irradiated mice (2 x het, 2 x homo). I think these results are not statically compelling (1-2 mice out of 25).

Response:

We agree with the reviewer. We detected the aberrant population in 3 of 75 mice without irradiation (4%) and in 4 of 27 mice with irradiation (15%). Although the results are not significant, we observe a trend. Our results suggest that the variant has a low predisposition to the development of malignant cells, and that its frequency increases when animals are exposed to irradiation, and thus to the development of second events.

To obtain significant results, a larger cohort of mice would be required. However, our study complies with the 3R principle in animal experimentation (reduce, replace, refine) as approved by our Bioethics Committee of the University of Salamanca. A total of 25 mice per genotype for a 2-year study is a high and appropriate number of mice and is in accordance with what has been described in the literature, where study groups often have between 10-30 mice.

These data suggest that the presence of the aberrant cell population largely is independent of the L43S mutation.

Response:

Thanks to the reviewer for the comments. The aberrant Mac1+ Sca1+ ckit- cells were not found in any WT mice, with or without irradiation. These results suggest that the presence of the population is dependent on the p.Leu43Ser variant. However, as previously mentioned, it has a low causality (4%, without irradiation, and 15% with irradiation).

This is partly confirmed by the RNA seq results which showed dysregulation of multiple genes only in mice with aberrant populations, but not in all L43S/L43S animals.

Response:

We thank the reviewer for this observation. Germline mutation per se in RUNX1 does not cause the disease, as acquisition of second events is required. It is therefore comprehensible that heterozygous and homozygous mice that have not acquired second events (and thus developed the disease) have a transcriptome similar to that of healthy WT mice.

We have rewritten the sentence as it was previously difficult to understand and could lead to confusion (lines 248-253).

Altogether, the presented data suggest that in animals with aberrant cell population a second mutagenic event has occurred that has led to the development of the observed phenotype, but the L43S mutation has little effect on it. This is partly confirmed by the lack of dramatic AML phenotype in homozygous mice versus wild type mice and by the fact that corresponding L56S mutation in human is considered benign (see discussion).

Response:

We agree with the reviewer that the p.Leu43Ser variant has little effect on disease progression. It is also discussed in the discussion section, as mentioned by the reviewer.

Another issue is that this second mutagenic event is unknown. Moreover, these second hits could be different in all mice with aberrant populations thus hampering the interpretation of results. Taking into account the lack of a dramatic phenotype in most of the studied animals one might assume that the development of the aberrant population depends on the second event, while Runx1 mutations only marginally contribute to this phenotype. This is exemplified by the higher percentage of aberrant population was higher in non-irradiated homozygous mice versus heterozygous (Fig1) suggesting that the L34S/L34S genotype makes mice prone to the development of this aberrant population, but still without second event there will be no aberrant population. I think that it is important to identify this second hit event in affected mice and study its relationship with Runx1 mutations.

Taking together the complex relationship between Runx1 mutations and possible second hit mutations should be investigated in greater details.

Response:

We thank the reviewer for highlighting this important issue. As mentioned, second events are required for the development and proliferation of the aberrant population, and we cannot rule out that it might be different in each affected mouse. Considering that somatic mutation of the other allele of RUNX1 is the most frequent mutation, we analyzed all exons of RUNX1 in the mice with the aberrant population by Sanger sequencing, but did not see any somatic mutation with a VAF>5.

Since many other second events are known (ASXL1, TET2, TP53, etc) it is very difficult to do a targeted screen. This could be solved with a gene panel, but it is important to note that this is a murine model, and a commercial one cannot be used (since they are designed for human samples) so it is very difficult to carry out that study. In addition, considering that genes for DNA repair, apoptosis, etc. are deregulated (based on the RNA-seq data), a commercial myeloid panel would not be suitable either, since it does not contain these types of genes. The solution would be an exome, but we do not have enough sample, and its analysis is too complex for us to carry out.

Minor issues

The most common second event that leads to the development of AML is the mutation of the second unaffected RUNX1 allele. But L65S mutation seems to be benign in human so why this particular mutation was selected for analysis?

Response:

We thank the reviewer for the question. As we mentioned in the previous question, we conducted the Sanger sequencing analysis of all the exons of RUNX1 in the affected mice to detect somatic mutations in the gene, but we did not find any candidate variant, so we can say that the somatic mutation of RUNX1 is not the second event that triggers the disease.

On the other hand, we wanted to study the L56S variant because, despite being considered as benign, we see that many patients with thrombocytopenia (and in specific cases, with AML) only have this candidate variant after molecular analysis, so we wanted to demonstrate its role and thus confirm its low association with the disease.

It is important to note that, although these are monogenic diseases, the role of other disease-modulating genes (polygenic risk score) is becoming increasingly known. Therefore, although the L56S variant is a polymorphism, we cannot rule out that it has a modulatory effect (together with the presence of mutations in other genes that are not per se disease-causing).

The changes in mRNA levels of RNA-seq identified dysregulated genes should be confirmed in independent experiments using qPCR for these genes (e.g. for Rapgef1 gene).

Response:

We appreciate the reviewer's suggestion. However, we are unable to perform the study. All the RNA from the mice was spent on the RNA-seq assay. Considering that the sample is obtained upon animal sacrifice, we cannot have more samples to perform the study. It is worth mentioning that we tried to do an ICC on BM parafilm samples, but the technique does not work properly, so unfortunately, we cannot validate that result.

Fig1AB- Experimental points are not labeled (those that are shown in green, red and blue). Fig1 title should be in bold.

Response:

We apologize to the reviewer. The labels are at the bottom of the figure. Considering that it can be confusing, we have moved the color labels just below the specific figure.

Reviewer 2 Report

Comments and Suggestions for Authors

The article is well-written, with a clear and detailed structure covering the introduction, methods, results, and discussion. 

There are this correction to do: 

  • line 72: how many male and female was used for each groups? The number of mice (25) may be insufficient to obtain statistically significant results, especially considering genetic and environmental variability.
  • line 73: lack the positive and negative control for the effect of 4Gy. How many male and female was used for each groups? there are the difference of the effect in male adn female mice? Also the number of mice may be insufficient to obtain statistically significant results.

Author Response

The article is well-written, with a clear and detailed structure covering the introduction, methods, results, and discussion. 

Response:

We would like to express our acknowledgment to this Reviewer for the positive feedback.

There are this correction to do: 

Line 72: how many male and female was used for each groups? The number of mice (25) may be insufficient to obtain statistically significant results, especially considering genetic and environmental variability.

Response:

We thank the Reviewer for pointing out this. The study included 37 females and 38 males, so there is gender equality. We included the information in the article (line 87-88).

We agree with the reviewer that the results are not significant, but we observe a trend. Our results suggest that the variant has a low predisposition to the development of malignant cells, and that its frequency increases when animals are exposed to irradiation, and thus to the development of second events.

To obtain significant results, a larger cohort of mice would be required. However, our study complies with the 3R principle in animal experimentation (reduce, replace, refine) as approved by our Bioethics Committee of the University of Salamanca. A total of 25 mice per genotype for a 2-year study is a high and appropriate number of mice and is in accordance with what has been described in the literature, where study groups often have between 10-30 mice.

Line 73: lack the positive and negative control for the effect of 4Gy. How many male and female was used for each groups? there are the difference of the effect in male adn female mice? Also the number of mice may be insufficient to obtain statistically significant results.

Response:

We thank the reviewer for the suggestions. We can consider WT mice as a negative control for the effect of 4Gys, since none of the mice have developed aberrant cells. However, obtaining positive irradiation control is more complex, since it should be an animal with a genetic modification that causes in the 100% of the cases a neoplasm after irradiation (for example, a mouse with a germline TP53 mutation), but unfortunately, we do not have this in the laboratory.

Regarding gender, 11 female mice and 16 male mice were irradiated, so there´s also an equality in gender. We included the information in the article (line 88-89).

Finally, concerning the number of animals, we consider that 9 per genotype is not a high number, but this study is a validation of the previous study without irradiation, where we clearly observed how the variant p.Leu43Ser has a role in the development of aberrant cells. This phenotype is exacerbated when mice undergo irradiation, due to the more rapid acquisition of second events promoting the disease.

Reviewer 3 Report

Comments and Suggestions for Authors

The manuscript “Examining the effects of the RUNX1 p.Leu43Ser variant on  FPD/AML phenotypes using a CRISPR/Cas9-generated knock-in murine model.” This manuscript aimed to examine the role of the variant by identifying an abnormal cell population in two homozygous mice and one heterozygous mouse carrying the variant. Additionally, these mice exhibited atypical expression of genes related to apoptosis and DNA repair. However, below mentioned points should be improved.

Comment: Add the significance of myeloid leukaemia in the introduction part of the manuscript with recent references in a short paragraph.

Line 82-100: Is this your lab protocol? Add the reference

Line 94-100: Add more details with the reference/s

Comment: Add the “Ethics statement” at the start of the Materials and Methods section (after Line 68).

Line 231-234: Rephrase the sentence: “We first compared three non-irradiated unaffected mice carrying the variant (one RUNX1WT/L43S and two RUNX1L43S/L43S) vs. three non-irradiated RUNX1WT/WT, and we observed only 5 dysregulated genes (Gm16698, Vκ21G, Rbm44, Gzma, and the LncRNA 5830416I19Rik), confirming that the variant p.Leu43Ser per se is not responsible for the phenotype”.

Line 362: Add more details on future perspectives

Comment: What is the difference between the study and comparison with other studies in the same direction? What are the novelty points?

Comment: Check the references following the journal format.

Author Response

The manuscript “Examining the effects of the RUNX1 p.Leu43Ser variant on  FPD/AML phenotypes using a CRISPR/Cas9-generated knock-in murine model.” This manuscript aimed to examine the role of the variant by identifying an abnormal cell population in two homozygous mice and one heterozygous mouse carrying the variant. Additionally, these mice exhibited atypical expression of genes related to apoptosis and DNA repair. However, below mentioned points should be improved.

Add the significance of myeloid leukaemia in the introduction part of the manuscript with recent references in a short paragraph.

Response:

We thank the reviewer for the commentary and the improvement in the introduction of the article. We have included a small paragraph highlighting the significance of the neoplasm and the treatment of patients:

“Therefore, an accurate and early diagnosis of these patients is critical for their prognosis and appropriate clinical management. Patients with FPD/AML require bone marrow examination and close follow-up for early recognition of transformation to neoplasm and, subsequent, inclusion into a hematopoietic progenitor cell transplanta-tion program [9].” (line 58-62)

Line 82-100: Is this your lab protocol? Add the reference.

Line 94-100: Add more details with the reference/s

 Response:

We thank the reviewer for the suggestion. It is a specific protocol of the laboratory adapted from the bibliography. The reference has been included (ref. 20, line 95 and 103).

Comment: Add the “Ethics statement” at the start of the Materials and Methods section (after Line 68).

Response:

The suggested change has been done.

Line 231-234: Rephrase the sentence: “We first compared three non-irradiated unaffected mice carrying the variant (one RUNX1WT/L43S and two RUNX1L43S/L43S) vs. three non-irradiated RUNX1WT/WT, and we observed only 5 dysregulated genes (Gm16698, Vκ21G, Rbm44, Gzma, and the LncRNA 5830416I19Rik), confirming that the variant p.Leu43Ser per se is not responsible for the phenotype”.

Response:

 We apologize to the reviewer for the difficulty understanding the sentence. It has been rewritten (lines 248-253).

“In the initial study, a comparative analysis was conducted on three non-irradiated, unaffected mice carrying the variant (one RUNX1WT/L43S and two RUNX1L43S/L43S) versus three non-irradiated RUNX1WT/WT mice. The results of this analysis revealed only five dysregulated genes (Gm16698, Vκ21G, Rbm44, Gzma and the LncRNA 5830416I19Rik). These findings indicate that the variant p.Leu43Ser does not, in itself, cause the observed phenotype.”

Line 362: Add more details on future perspectives

Response:

 We thank the reviewer for the suggestion. As required, we add some information about future perspectives at the end of the conclusions (lines 374-377):

“However, these results should be expanded and confirmed by studying patients and more variants in the RUNX1 gene. In this context, it would be of significant interest to examine specific genes regulated by RUNX1, as the measurement of them could provide valuable insights into the pathogenicity of the variant.”

Comment: What is the difference between the study and comparison with other studies in the same direction? What are the novelty points?

Response:

We thank the reviewer for this interesting question. In this work we demonstrate the pathophysiological role of a variant in a model without other mutations. Patients with IT carrying the variant have been reported in the literature, but we did not know the causality or its involvement in the disease. Although cellular studies have been done with the variant (Decker, M., et al. Leukemia 2021 - reference 15 in the manuscript), they have not been done in an animal model, which more faithfully reproduces the disease.

Comment: Check the references following the journal format.

Response:

Thank you for the constructive suggestions. The references have been updated to the journal format, and we have also checked the English.

Round 2

Reviewer 1 Report

Comments and Suggestions for Authors

Authors have carefully replied to my comments. Nevertheless, none of the experimental issues have been addressed. The presented study was performed at the appropriate technical level, but the obtained results are somewhat preliminary and require further investigation. Therefore, in the manuscript all the limitations  of the presented study (that were discussed in the comments) should be clearly stated and thoroughly discussed.

Author Response

As requested by the reviewer, the information has been included in the manuscript (highlighted in yellow)

Reviewer 3 Report

Comments and Suggestions for Authors

Ok.

Author Response

Thanks to the reviewer for improving the quality of the article.